# Optimal Enzymatic Hydrolysis of Sweet Lupine Protein towards Food Ingredients

**Diana Pasarin** [1] , **Vasile Lavric** [2] , **Cristina Emanuela Enascuta** [1,*] , **Andra-Ionela Ghizdareanu** [1,*] and **Catalin Bogdan Matei** [1]

1    National Research and Development Institute for Chemistry and Petrochemistry-ICECHIM, 202 Splaiul Independentei, 060021 Bucharest, Romania
2    Faculty of Chemical Engineering and Biotechnologies, University Politehnica of Bucharest, 1-7 Gheorghe POLIZU St., Sector 1, 011061 Bucharest, Romania
*    Correspondence: cristina.enascuta@gmail.com (C.E.E.); ghizdareanuandra@gmail.com (A.-I.G.)

**Abstract:** Although its high protein content, bioactive compounds, and dietary fibers have many health benefits, lupine (*Lupinus* sp.) was undervalued as a resourceful plant. In the last years, however, the number of studies on the use of lupine as a potential food ingredient has increased. In this study, obtaining a sweet lupine protein concentrate (SLPC) hydrolysate with a high degree of hydrolysis (DH) and a simultaneous low intensity of bitterness was pursued. The experimental design was carried out, according to Taguchi methodology, using three experimental parameters: enzyme concentration (0.5–1.5%), temperature (50–60 °C), and time (1–4 h), with three levels each. The optimal conditions for the enzymatic hydrolysis process of SLPC with Alcalase 2.4 L were enzyme concentration 1.5%, temperature 50 °C, and time 4 h, for which the best DH, 41.96%, was achieved. The SLPC hydrolysate as a food ingredient was characterized in terms of DH, bitter taste intensity, amino acid profile, and techno-functional properties. The results showed an increase of water binding capacity to 1.45 g/g, emulsification activity of 50.91%, and 92% stability of the emulsion, while the evaluated intensity of bitterness was 4.6 on a 7- point scale. Based on its technological, functional, sensory, and chemical characteristics, this study recommends the use of SLPC hydrolysate as a food ingredient in various food matrices.

**Keywords:** sweet lupine protein; enzymatic hydrolysis; food ingredient; sensory analysis

## 1. Introduction

There is an increase in consumer demand for new, healthy ingredients that bring value to the food, which motivated the researchers to look up new alternative sources of non-genetically modified (NGM) and cheap vegetable protein [1]. Soybeans and peas are the most common vegetable protein sources, but there are other plants whose protein content is of interest, among them, sweet and bitter lupine. Lupine (*Lupinus* sp.) is an herbaceous plant from the family *Leguminosae* (*Fabaceae*), widespread in the Mediterranean part of Europe, Africa, America, and Australia. The leaves are palmate and dark green. The plant forms clusters of flowers at the top of the stem and has a wide range of colors: white, yellow, blue, lilac, purple, and yellowish orange. The fruit of the lupine is called a pod and contains about five seeds called lupines [2].

Despite its high protein and dietary fiber content, lupine is an undefined plant with potential health benefits. Lupine has good adaptability to extreme environmental conditions, and therefore, cultivating this plant can be done in a sustainable way [3]. Despite the potential of lupine to become a unique and healthy food ingredient for humans, it is frequently used as animal feed. Its poor sensory quality (bitter lupine) prevents lupine usage as a food ingredient. However, due to the growing demand for both healthier and NGM ingredients, and sustainably produced foods, an increase in the number of studies on its potential as a food component has emerged.

Lupine flour and seeds can be used in food as a nutritional source of vegetable protein, such as soy (high protein and low starch content). In addition to macronutrients, they are rich in polyphenols, carotenoids, and phytosterols, offering several health benefits [4].

Sweet lupine is preferred over bitter ones to produce protein concentrates and isolates (PC&I) for use in the food industry because it has a low content of quinolizidine alkaloids, which are responsible for the bitter taste [5]. PC&I are mainly used for the development of nutritional foods [6,7], while protein hydrolysates are used in the food industries to obtain food ingredients [8,9].

The SLPC is rich in bioactive peptides associated with health benefits, also having similar techno-functional properties to proteins from other sources [10–13]. In the process of digestion, protein hydrolysates are more easily assimilated than PC&I, bioactive peptides being better absorbed at the level of the intestinal mucosa, entering the bloodstream directly, therefore keeping their bioavailability. Consequently, SLPC can be used as a food ingredient [14–16].

Bioactive peptides are dormant in their parent protein sequences, but they can be activated when released by various hydrolysis methods. Enzymatic hydrolysis is the most efficient and safe to produce the bioactive peptides [17,18]. The specificity of the enzyme is decisive during the hydrolysis process for the number of such peptides that are obtained [19]. In terms of techno-functional properties, the enzymatic protein hydrolysates can contribute to the water retention capacity increase, texture, and gelling of food products, due to their good foaming and emulsifying properties, including good solubility in a wide pH range.

The most common proteolysis indicator used to determine the cleaved peptide bonds in a hydrolysis process is DH. It represents the proportion (%) of cleaved bonds and evaluates the efficiency of protein–peptide bond leakage [20]. Han and Ren [21] used Alcalase 2.4 L for the hydrolysis of corn protein obtaining the maximum DH value in the pH range of 8.0–8.50.

Shuai et al. utilized an optimal hydrolysis temperature of approximately 50 °C for 15 min and adjusted the pH of the aqueous solution containing pea protein to the optimal pH of the enzymes (Alcalase 2.4 L and Trypsin had a pH of 8; Neutrase and Flavourzyme had a pH of 7). The enzymatic hydrolysis degree for the four enzymes, from greatest to least, were Trypsin, Alcalase 2.4 L, Neutrase, and Flavourzyme based on the activity unit and specific restriction site of each respective enzyme. A similar result was reported for pea protein isolate treated with eleven proteolytic enzymes at different enzymatic hydrolysis times [22]. Islam, M., et al. (2022) also noted differences in yields for various protein hydrolysates and peptides during hydrolysis using eleven proteolytic enzymes depending on enzyme activity and temperature. Hydrolysis parameters, such as digesting duration, pH, temperature, buffer-to-substrate ratio, and enzyme-to-substrate ratio (E/S), significantly impacted amino acid profiles, functional characteristics, and antioxidant properties. The soy protein hydrolyzed best at a temperature of around 50 °C. Under ideal conditions, the average hydrolysate yields from the freeze-dried soybean protein were closely correlated with the DH, with Protamex producing the greatest yield (19.77%) and Alcalase 2.4 L the second highest (16.08%) [23].

In addition, the saturation of DH could be due to the small concentration of peptide bonds available for hydrolysis, enzyme inhibition, or enzyme denaturation [23].

However, the use of enzymatic protein hydrolysates in food is limited, and in relatively low concentrations, owing to their intrinsic strong bitter taste that occurs after hydrolysis.

The latter is associated with the type of enzymes used for hydrolysis, together with the hydrophobicity, DH, molecular weight, and amino acid sequence obtained after the process [24].

The intensity of the bitter taste is enhanced with the increase of DH, due to the breaking of the protein bonds, which exposes the hydrophobic amino acid residues [25].

Traditional debittering methods of protein hydrolysates are absorption of bitter peptides on activated carbon, chromatographic removal of different matrices, and selec-

tive extraction with alcohols, which are procedures that lead to the loss of amino acid residues from hydrolysates, thus reducing the techno-functional properties of these types of hydrolysates [26]. The type of enzyme and the conditions (time, temperature, pH, and enzyme concentration) in which the hydrolysis is carried out, lead to the appearance of peptides with different degrees of hydrophobicity with a direct influence on the techno-functional properties [27].

The main purpose of this research was to simultaneously obtain a high DH of sweet lupine protein and a low level of bitterness, thus preventing the further use of any procedure to reduce the latter. Another goal was to quantify the techno-functional properties of the SLPC and hydrolysate.

A permanent comparison with the literature was done to verify if the obtained enzymatic protein hydrolysates qualify as food ingredients.

The optimization of the enzymatic process to find the experimental conditions that maximize the main goal, as well as fulfilling the second goal, was performed using the Taguchi design of experiment methodology [28,29].

## 2. Materials and Methods

SLPC as a yellow powder (*Lupinus angustifolius* L., 55% protein content, fat 9.8%, carbohydrates 7.6%, dietary fibers 14.6% FRALU-CON, 12760010137, FRANK Food Products) was used in all experiments. Food-grade enzyme Alcalase 2.4 L (P4860) (Protease from Bacillus licheniformis) was obtained from Sigma–Aldrich.

The reagents (heptafluorobutyric acid, Sigma–Aldrich; acetonitrile—MS grade, Agilent; ultrapure water—LC/MS grade, Agilent; and amino acids standards, Sigma–Aldrich kit) used for the chromatographical characterization were of analytical grade available commercially.

All solutions for the sensory analysis were prepared with distilled water and caffeine ingredient (Dohler, Germany).

### 2.1. Experimental Design via Taguchi Methodology

Considering the manufacturer's recommendations for the Alcalase 2.4 L (the pH range 7–9, the temperature range from 30 °C to 65 °C) and the protein enzymatic hydrolysis studies carried out with this enzyme, the following parameters were chosen for SLPC hydrolysis: temperatures ranging from 50 °C to 60 °C, time of hydrolysis from 1 to 4 h, and the concentration of the enzymes from 0.5% to 1.5%.

The optimization study for the enzymatic hydrolysis of SLPC used the following experimental parameters: enzyme concentration, temperature, and time, with three levels each, as presented in Table 1.

**Table 1.** Parameters and their levels.

| Levels | Parameters | | |
|--------|------------------------------|-----------------------|------------------|
| | **(A) Enzyme Concentration (%)** | **(B) Temperature (°C)** | **(C) Time (Hours)** |
| (1) | 0.5 | 50 | 1 |
| (2) | 1 | 55 | 2 |
| (3) | 1.5 | 60 | 4 |

The Taguchi design methodology, applied as implemented in the MINITAB-18 software (Minitab, LLC, State College, Pennsylvania, USA), involves defining an objective or target for a process performance measure, such as a DH and/or level of bitterness. Design parameters are then determined, and their levels of variation are specified to form an orthogonal array. The experiments indicated in the array are conducted to collect data on the performance measure and then, finally, the data is analyzed to understand how the parameters impact the performance measure. Taguchi's orthogonal array experimental design allows for the examination of many parameters' effects on a performance measure in a condensed set of experiments. Determining the levels of the variables requires an understanding of the process and parameters, with the range of parameters dictating how

many values can be tested and how far apart they should be. The appropriate orthogonal array can be selected by using the array selector table, which provides the necessary information to find the name of the array, and where to find it. This array, created with the Taguchi algorithm, ensures that each variable and setting are tested equally.

For three factors and three levels, nine experimental conditions were generated, according to the Taguchi methodology, as presented in Table 2.

**Table 2.** Experimental design for the SLPC hydrolysis.

| Run | A | B | C |
|-----|-----|-----|-----|
| R1 | 0.5 | 50 | 1 |
| R2 | 0.5 | 55 | 2 |
| R3 | 0.5 | 60 | 4 |
| R4 | 1 | 55 | 2 |
| R5 | 1 | 60 | 4 |
| R6 | 1 | 50 | 1 |
| R7 | 1.5 | 60 | 4 |
| R8 | 1.5 | 50 | 1 |
| R9 | 1.5 | 55 | 2 |

### 2.2. Enzymatic Hydrolysis of SLPC

Enzymatic hydrolysis of SLPC was carried out in a batch reactor. All experiments were run in identical Erlenmeyer flasks (250 mL), using the same amount of 12.5 g of SLPC with 55% protein concentration, mixed with distilled water in a mass ratio of one part protein to ten parts water. The mixtures were homogenized at 10,000 rpm for 2 min using T18 digital Ultra Turrax, IKA, and then heated to 37 °C. After reaching this temperature, the pH was adjusted to 8, which is optimum for the enzyme Alcalase 2.4 L (P4860). The enzymes were added to the mixture in different concentrations, according to the orthogonal experimental design (see Table 2).

Hydrolysis was performed in a Nahita 639/70 incubator at the designed temperatures and for the designated periods, according to Table 2; the orbital shaker was set to 160 rpm. The pH value was checked and adjusted every hour. The process was stopped at the end of the designated hydrolysis time, and then, the samples were heated up to 95 °C for 10 min to inactivate the enzymes. After that, the samples were cooled on ice and neutralized, and the obtained hydrolysates were centrifuged at $5500 \times g$ for 30 min at 4 °C (to separate the unhydrolyzed residue from the soluble hydrolyzed material). The supernatants were collected, concentrated on a Heidolph Hei-VAP Core HL G3 rotary evaporator, and freeze-dried in an ALPHA 1–2 LD plus, Martin Christ freeze-dryer.

The lyophilizates were stored under a vacuum and kept at −20 °C until the analyses were performed. All experiments were performed in duplicate.

### 2.3. The Degree of Hydrolysis

The DH was determined using the method described by Kaewka et al. [30]. The solubility index in trichloroacetic acid (TCA), also called the non-protein nitrogen method, consists of the determination of soluble nitrogen after precipitation with TCA. An amount of 10 mL protein hydrolysate was mixed with 10 mL 10% TCA, which was allowed to stand for 30 min, and then centrifuged at 4 °C for 10 min at $6700 \times g$. The nitrogen content of the supernatant and from the sample is determined by the Kjeldahl method. Each hydrolysate was analyzed in duplicate, and DH is calculated using the Equation (1) [31]:

$$\text{DH(\%)} = \frac{\text{total nitrogen soluble in TCA and present in supernatant}}{\text{total nitrogen in the substrate sample}} \times 100 \tag{1}$$

This method does not determine the number of broken peptide bonds but measures TCA-soluble nitrogen, which is found only in small amino acids and peptides. This method precipitates the unhydrolyzed proteins, which are still present [32].

*2.4. Sensory Evaluation—Scaling Method*

Sensory evaluation for the bitter taste of hydrolysates was conducted according to the method described by Lawless and Heymann [33]. The scaling method (intensity of taste attribute) used a bitter standard solution and a trained panel formed by 5 women and 5 men aged between 25 and 30 years old. The training of the panel was made by using 7 bitter standard solutions representing a point scale from 1 to 7 (1—no bitterness; 2—weakly bitter; 3—mildly bitter; 4—bitter; 5—medium bitter taste, 6—strongly bitter; 7—extreme bitterness) consisting of various concentrations of caffeine in water: 0%, 0.5%, 1%, 1.5%, 2%, 2.5%, and 3.5% [34].

A 7-point hedonic test is one of the most reliable methods for evaluating a food product in terms of its sensory characteristics. The seven points give a good overall picture of how a consumer would perceive the product and can provide valuable insights into the product's marketability. Using this method, researchers can produce accurate and reproducible results, allowing them to make informed decisions about how to improve their product before launch. A 7-point hedonic test is generally preferred for sensory analysis because it provides a simpler and more reliable method of evaluating a food product.

The 7 points are easier to remember and less likely to be misinterpreted, resulting in more accurate and consistent results. Additionally, the 7-point scale is better suited for wider use since it can provide more information with fewer points. A 7-point hedonic test is one of the most reliable methods for evaluating a food product in terms of its sensory characteristics. The seven points give a good overall picture of how a consumer would perceive the product and can provide valuable insights into the product's marketability. Using this method, researchers can produce accurate and reproducible results, allowing them to make informed decisions about how to improve their product before launch.

The main difference between a 7-point hedonic test and a 9-point hedonic test is the number of points used in the assessment. A 7-point hedonic test is designed to measure three distinct aspects of sensory perception (appearance, aroma, and flavor). In contrast, a 9-point hedonic test evaluates five different aspects of sensory perception (appearance, aroma, flavor, texture, and intensity). This gives researchers a more comprehensive view of how they can improve the overall product.

The 7-point hedonic test is more suited for assessing the overall level of bitterness in a raw sample, while the 9-point hedonic test is more advantageous for determining the intensity of bitter notes that can be masked by a food matrix [35–37].

Different studies of sensory analyses were conducted with a maximum of 2% $w/w$ sweet lupin protein hydrolysate. After further testing in the lab, a panel sought to assess the acceptability of the hydrolysate at different concentrations (1–5%) in water. Eventually, it was agreed that the maximum acceptability threshold for this hydrolysate in water is 3.5% [38,39].

The panelists were asked to taste the bitter standard solutions (the samples were labeled with the intensity number) in order from lowest to highest and rinse with water in their mouths between each sample. Panelists then assigned a score of bitterness according to the point scale to each hydrolysate sample (labeled with random three-digit codes). The raw SLPC and each hydrolysate sample were tested by panelists at a concentration of 3.5%. The results were analyzed by commuting means, standard deviations, and standard error of the mean. For each result, we performed a simple one-way ANOVA analysis [40].

*2.5. Techno-Functional Properties*

2.5.1. Water Binding Capacity (WBC)

Water binding capacity (WBC) was determined according to the slightly modified method described by Rodriguez-Ambriz et al. [41]. An amount of 0.5 g of sample is mixed with 5 mL of distilled water in a centrifuge tube, homogenized using a vortex for 5 min, and incubated at room temperature for 30 min. After centrifugation at $5500 \times g$ for 20 min, the supernatant is decanted and the mass of the tube with the precipitate is measured. The

WBC of each sample was determined in duplicate. The WBC is expressed by distilled water (g)/1 g of sample, according to Equation (2):

$$\text{WBC} = \frac{\text{weight of tube and precipitate (g)} - \text{weight of tube and dry sample (g)}}{\text{weight of dry sample (g)}} \quad (2)$$

### 2.5.2. The Emulsification Activity (AE)

The emulsification activity (AE) is determined according to the method presented by Fekria et al. [42] and Thakur and Nanda [43], with modifications. A 2% (*w/v*) protein hydrolysate solution is prepared and vortexed at low speed for 1 min. From the dispersion formed, 30 mL is taken and mixed with 30 mL of pure sunflower oil. The mixture is homogenized in an electric blender for 5 min and then centrifuged at $3500\times g$ for 10 min. The mixture is collected in a graduated cylinder and left for a few minutes until the emulsified layer becomes stable. The height of the emulsified layer and the total height of the graduated cylinder content are measured. The AE is calculated according to Equation (3):

$$\text{AE (\%)} = \frac{\text{height of the emulsified layer}}{\text{total height of the tube content}} \times 100 \quad (3)$$

### 2.5.3. The Stability of the Emulsion (SE)

The stability of the emulsion (SE) is determined after centrifugation of the previously prepared emulsion, followed by heating to 80 °C for 30 min, cooling to 15 °C, and centrifugation for 5 min. The emulsion is collected in a graduated cylinder and then held for a few minutes to stabilize the emulsified layer. The SE for each sample was determined in duplicate. Equation (4) is used to calculate the SE value:

$$\text{SE(\%)} = \frac{\text{height of the emulsified layer after heating}}{\text{total height of the tube content}} \times 100 \quad (4)$$

### 2.6. Identification of the Molecular Mass Distribution

The molecular mass distribution of peptides after hydrolysis was identified chromatographically and was performed with a high-performance liquid chromatography system coupled with a mass spectrometer analyzer and a time-of-flight detector, model 6224 TOF LC/MS (Agilent Technologies Inc., Santa Clara, CA, USA). System calibration was performed with mass reference solution (ESI-L Low concentration tunning mix, code G1969-85000, Agilent Technologies).

Peptide detection was performed using an orthogonal TOF mass spectrometer coupled to an ESI source with double spray needles for continuous perfusion of the reference mass solution. The drying gas was heated to 350 °C, with a flow rate of 9.0 L/min nitrogen at a pressure of 2.72 atm, and was used to desolvate the solution droplets. The spraying was induced with a capillary voltage of 3500 V, and the fragmentation voltage was 100 V.

Five microliters of the sample were filtered through a 0.22 μm PTFE syringe filter and injected directly into the MS, without chromatographic separation. The mobile phase, consisting of 80% acetonitrile with 0.01% trifluoroacetic acid, was used for sampling and transfer of the sample, using the quaternary HPLC pump, with a flow rate of 0.3 mL/min. The acquisition of data and their qualitative processing were carried out with the Mass Hunter software version B.04.00. The data acquisition range was 600–2000 $m/z$, set to 9894 scans and 1 scan/s. The total ion chromatogram and mass spectrum were recorded in the selected range. From the analyzed complex mixture, the signals with a specific molecular mass of amino acids, dipeptides, and oligopeptides were extracted.

### 2.7. Statistical Analysis

One-way analysis of variance (ANOVA) of the data was conducted using a General Linear Model procedure (for a statistically significant level of $p < 0.05$), as implemented in MINITAB-18 software.

## 3. Results and Discussions

### 3.1. The Degree of Hydrolysis

The values obtained for DH, after the determination of the total nitrogen content from supernatant and precipitation with TCA, are presented in Table 3:

**Table 3.** The results obtained for the nitrogen content and DH.

| Run | Nitrogen Content from the Supernatant (%) | DH (%) |
|---|---|---|
| R1 | 0.067 ± 0.008 | 6.03 ± 0.76 |
| R2 | 0.080 ± 0.010 | 7.20 ± 0.95 |
| R3 | 0.069 ± 0.003 | 6.27 ± 0.29 |
| R4 | 0.163 ± 0.008 | 14.74 ± 0.81 |
| R5 | 0.094 ± 0.008 | 8.52 ± 0.80 |
| R6 | 0.273 ± 0.005 | 24.61 ± 0.41 |
| R7 | 0.191 ± 0.004 | 17.23 ± 0.28 |
| R8 | 0.364 ± 0.005 | 32.80 ± 0.47 |
| R9 | 0.450 ± 0.010 | 40.54 ± 0.92 |

All results were expressed as mean value ± standard deviation of at least two measurements (n = 2).

The DH values obtained after SLPC hydrolysis with Alcalase 2.4 L ranged between 6.03 ± 0.76% (R1, see Table 2 for conditions) and 40.54 ± 0.92% (R9, see Table 2 for conditions).

### 3.2. Sensory Analysis

The trend of bitterness for the SLPC hydrolysates, of different degrees of hydrolysis, are presented in Figure 1.

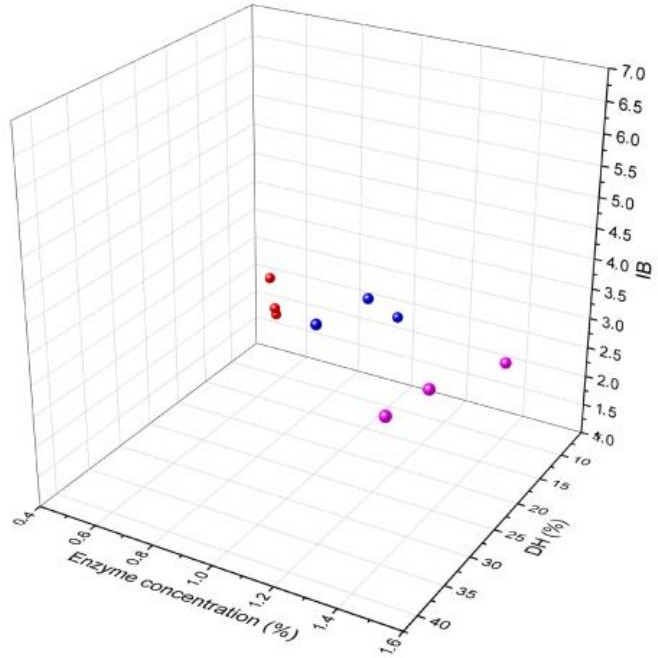

**Figure 1.** The influence of enzyme concentration on the IB and DH: red (0.5% enzyme concentration); blue (1% enzyme concentration); pink (1.5% enzyme concentration).

The proteins from vegetable sources or peptides with high molecular mass do not exhibit a bitter taste because their hydrophobic amino acids are oriented toward the interior of the molecule. In enzymatic hydrolysis, the hydrophobic amino acids are exposed depending on how the bonds between the molecules are broken during the process and how far the latter reaches.

As the DH increases, the bitter flavor becomes more intense because the exposed hydrophobic amino acid residues result from broken protein bonds [25]. Therefore, the aim

was to achieve a DH as high as possible so that consumers would accept the intensity of the bitter flavor. The intensity of bitterness for the SLPC and its hydrolysates was evaluated on a 7-point scale and the results are shown in Figure 1. The SLPC was evaluated with an intensity of bitterness of 1.5. The intensity of bitterness of the SLPC hydrolysates, except for R8 and R9 with a bitterness score of 3.70 and 3.90, was significantly higher than the value obtained for SLPC (for $p < 0.05$, when doing Tukey pairwise comparisons), but below the maximum of the 7-point scale. The SLPC hydrolysates with the highest DH value (40.54%) had a higher intensity of bitterness (3.90) than those with lower DH values. However, it did not exceed the maximum intensity limit established by the panel (of 5—strongly bitter) for the concentration at which these samples were analyzed. Schlegel et al. [12] analyzed a bitterness score of 7.2 and 5.7 (on a 10-point scale) for a lupine protein isolate hydrolysate (with a protein content of 92%) treated with Alcalase 2.4 L and the Alcalase 2.4 L + Papain, respectively. This may be due to the bitter peptide generation of Alcalase 2.4 L, which generally consists of hydrophobic amino acid residues [44].

### 3.3. Optimization of Hydrolysis Parameters by Taguchi Methodology

To obtain the optimal parameters for the enzymatic hydrolysis of SLPC, the experiments were conducted according to the Taguchi methodology. Based on the results obtained for the DH and the intensity of the bitterness, range, and signal-to-noise (S/N) ratio analysis, statistical analysis (ANOVA) was performed (Tables 4 and 5, and Figure 2).

**Table 4.** The range analysis for the DH and IB obtained for the L9 matrix.

| Run | Parameters | | | DH (%) | IB |
|---|---|---|---|---|---|
| | A | B | C | | |
| R1 | 0.5 | 50 | 1 | $6.03 \pm 0.76$ | $1.75 \pm 0.106$ |
| R2 | 0.5 | 55 | 2 | $7.20 \pm 0.95$ | $2.50 \pm 0.040$ |
| R3 | 0.5 | 60 | 4 | $6.27 \pm 0.29$ | $1.88 \pm 0.049$ |
| R4 | 1 | 55 | 2 | $14.74 \pm 0.81$ | $3.25 \pm 0.070$ |
| R5 | 1 | 60 | 4 | $8.52 \pm 0.80$ | $2.50 \pm 0.100$ |
| R6 | 1 | 50 | 1 | $24.61 \pm 0.41$ | $3.50 \pm 0.070$ |
| R7 | 1.5 | 60 | 4 | $17.23 \pm 0.28$ | $3.00 \pm 0.049$ |
| R8 | 1.5 | 50 | 1 | $32.80 \pm 0.47$ | $3.75 \pm 0.049$ |
| R9 | 1.5 | 55 | 2 | $40.54 \pm 0.92$ | $3.95 \pm 0.028$ |
| K (1)—DH | 6.50 | 21.14 | 12.66 | | |
| K (2)—DH | 15.95 | 20.82 | 16.17 | | |
| K (3)—DH | 30.19 | 10.67 | 23.80 | | |
| K (1)—IB | 2.05 | 3.00 | 2.67 | | |
| K (2)—IB | 3.09 | 3.23 | 2.92 | | |
| K (3)—IB | 3.56 | 2.45 | 3.10 | | |
| R—DH | 23.69 | 10.47 | 11.14 | | |

Data represent the means of three independent experiments. A—enzyme concentration (%); B—temperature (°C); C—time (hours); DH—degree of hydrolysis; IB—Intensity of bitterness; IB-ki = ΣIB at 'i' level in the same column/3; DH-ki = ΣDH at 'i' level in the same column/3; the difference between the highest and the lowest among K (1), K (2), K (3) is defined by the symbol "R".

**Table 5.** The analysis of variance (ANOVA) for the DH/(IB) obtained for the L9 (33) orthogonal matrix.

| Parameter | DF | Seq SS | Adj SS | Adj MS | F | *p* | IP (%) |
|---|---|---|---|---|---|---|---|
| The concentration of enzyme (%) | 2 | 247.836 | 247.836 | 123.918 | 102.95 | 0.010 | 77.06 |
| Temperature (°C) | 2 | 43.315 | 43.315 | 21.657 | 17.99 | 0.053 | 13.46 |
| Time (h) | 2 | 28.011 | 28.011 | 14.005 | 11.64 | 0.079 | 8.71 |
| Residual error | 2 | 2.407 | 2.407 | 1.204 | | | |
| Total | 8 | 321.568 | | | | | |

DF—Degree of freedom; Seq SS—Sum of squares; Adj SS—Adjusted sums of squares; Adj MS—Adjusted mean squares; F-value; *p*—value; IP—Influence of parameters on yield.

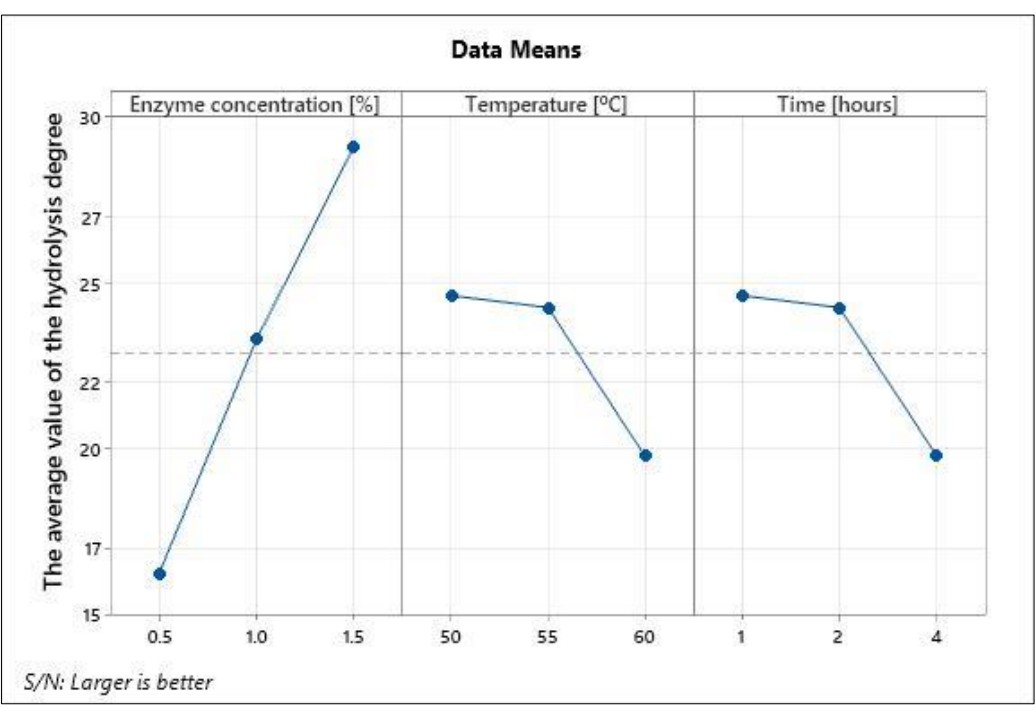

**Figure 2.** Main effects plot for S/N ratios.

### 3.3.1. Range Analysis

In statistics, the range analysis is the spread of results from the lowest to the highest value in the distribution, most commonly used to measure the variability, and the central tendency and to give descriptive statistics for summarizing the results set.

The range is calculated by subtracting the lowest value from the highest value; a large range means high variability, while a small range means low variability.

The range analysis shows the degree of influence of the parameters on the DH and their order is: A (concentration of enzyme) > C (time) > B (temperature). A (3) and C (3) have greater R-values and will have a greater influence on DH.

Considering that the IB was within the acceptable limit at the highest DH for R9 (maximum 5—strong bitter) at the analyzed concentration, for the other statistical analyses, only the values obtained for the DH will be considered.

### 3.3.2. S/N Ratio Analysis

Taguchi's methodology uses S/N ratios for assessing the effect of process parameters on the output of the process. The S/N ratios were calculated based on "larger is better" criteria as a higher hydrolysis degree is desirable. The effect of each parameter with their respective level is plotted in Figure 2.

DH and bitterness degree is most important in the enzymatic hydrolysis of SLPC. Therefore, from Tables 4 and 5, it could be seen that to obtain a maximum DH, the optimal conditions for the enzymatic hydrolysis process of SLPC with Alcalase 2.4 L are the following: enzyme concentration 1.5%, time 4 h, temperature 50 °C.

Considering the S/N ratio analysis on the DH presented in Figure 2, it is observed that this is directly proportional to the enzyme concentration and the duration of hydrolysis. In terms of temperature, in the range of 50–55 °C, the DH remains almost constant but decreases at temperatures higher than 55 °C.

### 3.3.3. Analysis of Variance (ANOVA)

The purpose of the ANOVA was to investigate which parameter significantly affected DH. Usually, the large F-value indicates which parameter has a significant effect (Table 5) [45].

The F-test on the three parameters indicated that A (3) is accepted as significant at the $p$-value ($p < 0.05$); however, B (2) and C (3) were not significant at a significance level of 0.05. Therefore, based on range analysis and ANOVA, the concentration of enzyme A (3) is the most significant hydrolysis parameter that influences the DH values.

By using the optimal parameters in MINITAB-18 software, the highest value of the DH is estimated at 43.78%.

The optimal conditions for the enzymatic hydrolysis process of SLPC with Alcalase 2.4 L (enzyme concentration 1.5%, time 4 h, temperature 50 °C) were experimentally verified and the SLPC hydrolysate was characterized in terms of the DH, bitter taste intensity, amino acid profile, and techno-functional properties.

The soluble nitrogen in the supernatant was determined by the Kjeldahl method. The value obtained, 0.4662%, was used to calculate the DH, 41.96%. After performing the sensory analysis with the panelists, the intensity of bitterness was evaluated at a value of 4.6 on the 7-point scale.

### 3.4. Techno-Functional Properties

The results obtained after determining the techno-functional properties of SLPC and SLPC hydrolysate are presented in Table 6.

**Table 6.** Techno-functional properties.

| Techno-Functional Parameters | SLPC | SLPC Hydrolysate |
|:---:|:---:|:---:|
| WBC (g/g) | 0.78 ± 0.021 | 1.45 ± 0.07 |
| AE (%) | 48.18 ± 0.14 | 50.91 ± 0.25 |
| SE (%) | 51.92 ± 0.25 | 92 ± 0.106 |

The SLPC hydrolysate had a WBC of 1.45 g water/g sample (Table 6). Karami and Akbari-Adergani [14] reported a water absorption of 1.16 mL/g for lupine protein isolate, a value comparable to that obtained for SLPC hydrolysate. Rodriguez-Ambriz et al. [41] compared the WBC of lupine protein (L. campestris) (1.7 mL water/g protein) with the value obtained for soy protein (2.2 mL water/g protein). After optimized hydrolysis, the hydrolysate doubled the WBC value compared with the results obtained for SLPC.

Amino acid residues, through their polarity, size, and shape, determine the interaction of protein molecules with polar molecules of water, influencing the ability to bind water. The higher the value of WBC, the better the food processing is in terms of shelf life and organoleptic properties [46].

The SLPC hydrolysate values for AE (50.91%) and SE (92%) after 30 min improved after hydrolysis, compared with the values obtained for SLPC (Table 6). The heat treatment applied during the enzymatic hydrolysis led to the exposure of the hydrophobic groups, initially inside the protein molecules. Proteins with high hydrophobicity also have a high emulsification activity [47].

The emulsifying property of proteins is influenced by surface hydrophobicity, solubility, molecular shape, and electrical charge. Protein molecules reduce the interfacial tension between the oily phase and the continuous aqueous phase, with the formation of a protective layer and the appearance of emulsions [12].

Some studies showed that vegetable proteins have the smallest emulsifying activity around the isoelectric point. Prima-Hartley et al. [48] showed that the most stable emulsions are those with smaller oil droplets (15–42 μm in diameter). According to Aluko et al. [49], by increasing the concentration of the sample, the size of the oil droplets will be smaller, and the emulsification activity will increase. On the other hand, a higher emulsifying activity may be due to a greater variety of peptide chains accessible for the formation of oil droplets. In addition, the presence of a higher quantity of sugars in the samples contributes to the increase of protein solubility and a better emulsification capacity.

### 3.5. Determination of the Molecular Mass Distribution

For SLPC hydrolysate, in the range of 85–250 $m/z$ (Figure 3), there are signals indicating hydrolysis up to specific amino acid fragments, respectively, 132 (leucine), 166 (phenylalanine), and 175 (arginine). The mass spectrum of the sample in the range 200–1500 $m/z$ (Figure 4) shows the presence of peptides with molecular weight from 203 (dipeptides) to 689 (oligopeptides).

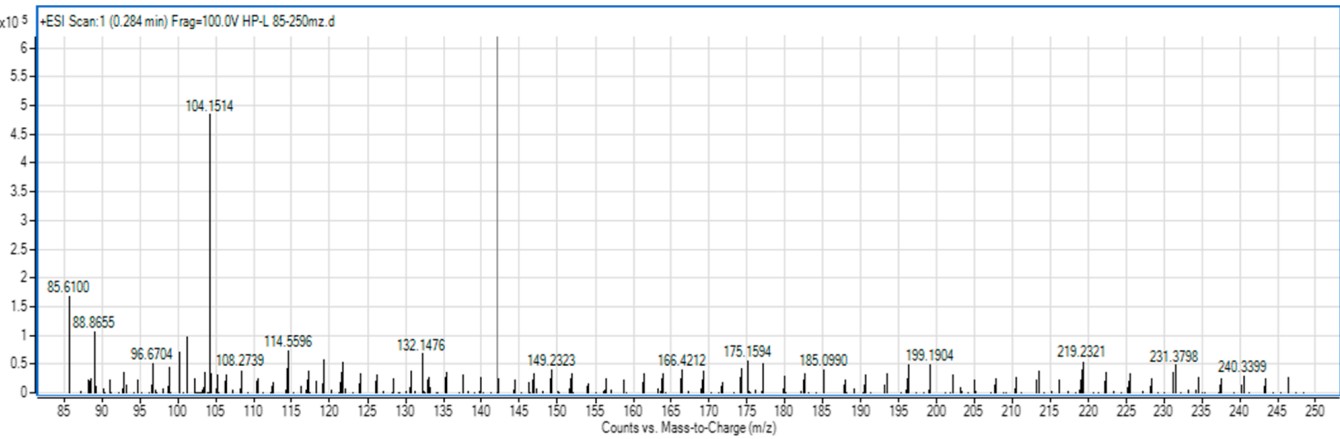

**Figure 3.** The spectrum of the hydrolysate sample in the domain 85–250 $m/z$.

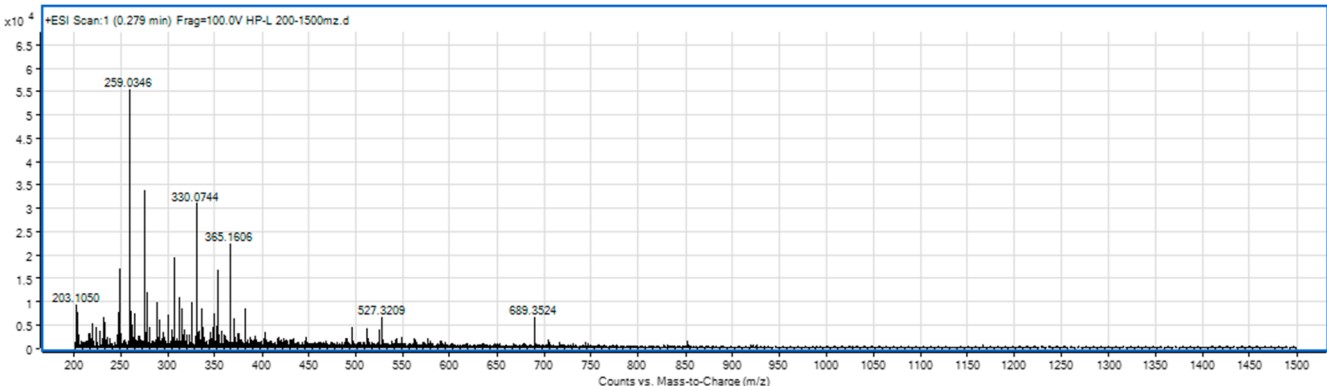

**Figure 4.** The spectrum of the hydrolysate sample in the domain 200–1500 $m/z$.

The activity of peptides is determined by the molecular weight, the number of amino acid residues, and their composition. Bioactive peptides are usually those that have short sequences of 2–20 amino acid residues and are inactive within the protein but can be released during digestion, processing foods, or through in vitro hydrolysis under the action of proteolytic enzymes [50]. The characteristic peak in the 85–250 $m/z$ range can be seen at 219 $m/z$, a mass range generally associated with dipeptides.

The most active peptides are the smallest, with chain length <7 amino acids or molecular weight <700–800 Da. Medium peptides (7–25 amino acids or 800–3000 Da) have significant bioactivity and are the easiest to identify with trypsin, many of them being unique for a certain protein [51]. The characteristic peak in the 200–1500 $m/z$ range can be seen at 259 $m/z$, a mass range generally associated with medium molecular weight dipeptides.

### 4. Conclusions

Optimization of SLPC was conducted to obtain a hydrolysate with both a high DH and a low level of bitterness, which could be used as a food ingredient.

The Taguchi experiment design methodology was used to find the optimal experimental conditions to maximize the DH. The effects of several parameters (enzyme concentration, temperature, and time) were studied on the SLPC hydrolysate properties. The optimal conditions for the enzymatic hydrolysis process of SLPC with Alcalase 2.4 L were enzyme

concentration 1.5%, temperature 50 °C, and time 4 h. The optimum SLPC hydrolysate was characterized in terms of techno-functional, sensory, and chemical properties. For SLPC hydrolysate with an optimum DH of 41.96%, the techno-functional properties such as WBC 1.45 $g/g$, AE 50.91%, and SE 92% were improved compared with the same properties of SLPC. After performing the sensory analysis with the panelists, the intensity of bitterness was evaluated at a value of 4.6 on the 7-point scale. The determination of the molecular mass distribution showed the presence of amino acid fragments, dipeptides, and oligopeptides with a molecular weight in the range of 130–700.

This study recommends the use of SLPC hydrolyzed as a food ingredient in different food matrices based on its techno-functional, sensory, and chemical properties.

## 5. Outlook

More complex analyses related to peptide profile to correlate it with sensory analysis of the SPLC hydrolysate in different food matrices, such as (in vitro digestibility with gastrointestinal simulants, stability in different food matrices, and sensory analysis) are ongoing. SLPC hydrolysate peptides have also been shown to possess a range of bioactivities, including antioxidant, anti-inflammatory, and antidiabetic properties. To accurately characterize the peptides for bioactivity, it will be necessary to analyze their molecular structure and primary sequence. It is also important to assess their stability to various environmental conditions, such as temperature, light, and pH. The SLPC hydrolysate will also be tested on different food matrices in different ratios/ concentrations in order to make it suitable for consumption. Taste and aftertaste acceptability tests will be conducted to determine the ideal matrix and quantity that customers prefer in terms of bitter taste detectability.

It should be noted that while SLPC hydrolysate can serve as a vegan alternative to dairy-based ingredients and as a protein source and flavor enhancer, it is not heat stable and should not be used for cooked dishes or heated foods. Processing at low temperatures is required to avoid any unwanted bitter flavor.

**Author Contributions:** Conceptualization, D.P., V.L. and A.-I.G.; methodology, D.P., V.L., A.-I.G. and C.E.E.; software, A.-I.G.; validation, D.P., V.L., A.-I.G. and C.E.E.; formal analysis, D.P., A.-I.G.; investigation, D.P., V.L., A.-I.G. and C.E.E.; resources, D.P., V.L., A.-I.G., C.E.E., C.B.M.; writing—original draft preparation, C.B.M.; writing—review and editing, D.P., V.L., A.-I.G. and C.E.E.; supervision, D.P., V.L. All authors have read and agreed to the published version of the manuscript.

**Funding:** This work was supported by a grant of the Romanian Ministry of Research and Innovation, CCCDI-UEFISCDI, project number Cod: PN-III-P3-3.5-EUK-2017-02-0035/contract 129/2019. This work was carried out through the PN 23.06 Core Program—ChemNewDeal within the National Plan for Research, Development and Innovation 2022–2027, developed with the support of Ministry of Research, Innovation, and Digitization, project no. PN 23.06.01.01.

**Institutional Review Board Statement:** Not applicable.

**Informed Consent Statement:** Not applicable.

**Data Availability Statement:** No new data were created or analyzed in this study. Data sharing is not applicable to this article.

**Conflicts of Interest:** The authors declare that they have no conflict of interest.

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
