# Peer review of "Optimal Enzymatic Hydrolysis of Sweet Lupine Protein towards Food Ingredients"

_fermentation, doi:10.3390/fermentation9030203_

Round 1
Reviewer 1 Report
The manuscript “Optimal enzymatic hydrolysis of sweet lupine protein towards functional ingredients for food industries” investigated the optimization of hydrolysis parameters of sweet lupine protein concentrate targeting low intensity of bitterness and high degree of hydrolysis. There are several major concerns about this work:
1. Content of this research failed to meet the scope requirements of the journal. The section this manuscript submitted to is industrial fermentation. However, the work is not related to fermentation or involving any microorganism activity.
2. Aim of this work was blur in title and abstract. The title should be revised to remove the phrase “functional ingredients”, which was misleading since the study did not work on any ingredients that are functional.
3. This work lacks sufficient novelty. Protein hydrolysis has already been widely acknowledged as an efficient way to improve protein functionality. It is also conventional to adjust parameters like time/temperature/enzyme quantity to achieve a better DH. Reducing of bitterness of the hydrolysate is an interesting topic, this work can further focus this point and characterize profile of the peptides.
4. Introduction should be more concise. Information at Lines 75-87 need to be integrated and organized. Literatures listed here is better to be from plant proteins, which might be better referenced.
Minor concerns:
1. Please add introduction about “Taguchi experiment design” in the manuscript.
2. Line 56 “SLPC” lacks the full name.
3. The proximate analysis data of SLPC should be provided, for example, protein content/lipid content…
4. Figure 1 is difficult to understand. The description in legend is not clear.
5. Line 372 “shows the presence of biopeptides with protonated molecular weight” is not accurate. Peptide with bioactivity is presumed but not a fact in the current study.
Author Response
Dear Reviewer,
We would like to thank you for your invaluable comments/suggestions/recommendations, helping us to improve our paper.
Please see the attachment.
Best regards,

Reviewer 2 Report
The manuscript submitted by Andra-Ionela Ghizdareanu et al. mainly dealt with optimal enzymatic hydrolysis of sweet lupine protein towards functional ingredients for food industries. Authors used enzymatic hydrolysis to get sweet lupine protein concentrate with a low bitterness. The paper was well written and organized. However, some problems or errors should be revised according to the following suggestions.
Equation 1, x should be x
line 176, what kind of standard solution ? Please make it clear.
line 178, why a point scale from 1 to 7 ? Normally we use 9-points as hedonic scale. Please make an explanation.
line 186, why a concentration of 3.5 g/L. please make an explanation.
Figure 2, please set the Y-axis maximum as 30.
Figure 3, please mark the characteristic peak of your peptide.
Figure 4, please mark the characteristic peak of your peptide.
please add the limitations and further research plan to your conclusions.
Author Response

(The authors gave the same response as above.)

Round 2
Reviewer 1 Report
I do not have further comments.
Reviewer 2 Report
Accept